# Ligand-Free Signaling of G-Protein-Coupled Receptors: Relevance to μ Opioid Receptors in Analgesia and Addiction

**DOI:** 10.3390/molecules27185826

**Published:** 2022-09-08

**Authors:** Wolfgang Sadee, John C. McKew

**Affiliations:** 1Cancer Biology and Genetics, College of Medicine, The Ohio State University, Columbus, OH 43210, USA; 2Pain and Addiction Center Laboratory, Department of Bioengineering and Therapeutic Sciences, UCSF, San Francisco, CA 94158, USA; 3Aether Therapeutics Inc., Austin, TX 78756, USA

**Keywords:** basal receptor signaling, GPCR, μ opioid receptor, opioid use disorder, dependence, naltrexone, 6β-naltrexol, morphine, etorphine

## Abstract

Numerous G-protein-coupled receptors (GPCRs) display ligand-free basal signaling with potential physiological functions, a target in drug development. As an example, the μ opioid receptor (MOR) signals in ligand-free form (MOR-μ*), influencing opioid responses. In addition, agonists bind to MOR but can dissociate upon MOR activation, with ligand-free MOR-μ* carrying out signaling. Opioid pain therapy is effective but incurs adverse effects (ADRs) and risk of opioid use disorder (OUD). Sustained opioid agonist exposure increases persistent basal MOR-μ* activity, which could be a driving force for OUD and ADRs. Antagonists competitively prevent resting MOR (MOR-μ) activation to MOR-μ*, while common antagonists, such as naloxone and naltrexone, also bind to and block ligand-free MOR-μ*, acting as potent inverse agonists. A neutral antagonist, 6β-naltrexol (6BN), binds to but does not block MOR-μ*, preventing MOR-μ activation only competitively with reduced potency. We hypothesize that 6BN gradually accelerates MOR-μ* reversal to resting-state MOR-μ. Thus, 6BN potently prevents opioid dependence in rodents, at doses well below those blocking antinociception or causing withdrawal. Acting as a ‘retrograde addiction modulator’, 6BN could represent a novel class of therapeutics for OUD. Further studies need to address regulation of MOR-μ* and, more broadly, the physiological and pharmacological significance of ligand-free signaling in GPCRs.

## 1. Introduction

Receptors have evolved to sense environmental and cellular stimuli with exquisite sensitivity. Often existing in large aggregates of proteins and other components, receptors respond to minute forces that trigger conformational changes, aggregate remodeling, and subsequent signaling processes. Maintaining receptors in their silent state requires intricate mechanisms to balance quiescence with responsiveness. If this restraint is incomplete, the receptor ‘leaks’, i.e., partially signals with ligand-free basal activity observed with numerous receptor classes [1,2,3]. Several factors can enhance basal signaling, for example, high receptor expression, mutations, including single nucleotide polymorphisms, membrane lipids, ions, and environmental conditions [4]. Pathophysiological consequences include neoplastic transformation occurring when mutations or overexpression activate basal signaling of growth factor receptors [4]. Numerous examples of basal signaling have been reported for G-protein-coupled receptors (GPCRs) [1,3], encoded by a large gene family. Enhanced basal signaling arising from activating mutations and overexpression of GPCRs has been linked to a variety of diseases, including cancer [5].

In contrast to causing pathogenic effects, basal activity can also serve physiological functions. For example, the ghrelin receptor, a GPCR, signals at ~50% of maximal capacity in a ligand-free form, leading to inverse agonists (antagonists that block ligand-free basal signaling) as treatment for eating disorders [6,7]. Therapy of psychoses involves inverse agonists of 5HT2A, also endowed with basal signaling [8]. Among opioid receptors, the δ opioid receptor (DOR) was the first discovered to display basal activity [9], followed by the μ opioid receptor (MOR) [10]. Assuming basal activity has a physiological role, one must ask whether and how basal activity is regulated—an area requiring more research. Another unresolved question asks whether agonists continue to bind to the activated receptor or dissociate after activation to generate ligand-free receptors, which could carry out part or all of the signaling, involving multiple downstream pathways. We know little about the factors impacting the equilibrium between resting and ligand-free active receptor states and whether or how ligands, including drugs, affect interconversion rates. This review will discuss functions and regulation of ligand-free signaling in MOR—an example potentially applicable to other GPCRs, including DOR.

## 2. Opioid Pain Therapy, Opioid Use Disorder, and Other Adverse Effects

Opioid analgesics are highly effective, but the risk of addiction, respiratory depression, bowel dysfunction, hyperalgesia, pruritis, cognitive impairment, and other adverse effects limit clinical utility. While OUD risk might be lower when analgesics are used therapeutically, for example, against cancer pain, a high risk of developing opioid use disorder (OUD) [11] has resulted in an epidemic of opioid overdose deaths [12] and millions in the US are non-medical or illicit opioid users [13]. Multiple strategies have been developed to treat or prevent OUD and adverse effects, some targeting the complexity of opioid receptor functions [14], but none have yet markedly curbed the opioid epidemic [13]. For example, positive allosteric MOR modulators [15] and ‘biased agonists’ acting via G proteins or β-arrestins [16,17,18] can cause analgesia with reduced adverse effects, as receptors exist in multiple conformations and signaling complexes that can be differentially activated [4]. Current OUD therapies (e.g., methadone, buprenorphine, or naltrexone maintenance) are effective but remain sub-optimal [19]. Effective therapies that break or prevent the vicious circle of OUD are urgently needed. We discuss here a novel approach to treatments of opioid dependence and other adverse effects, without blocking opioid analgesia nor causing withdrawal (previously presented in [20]). While known as an antagonist, 6β-naltrexol (6BN) appears to potently act as a selective ‘opioid addiction modulator’ at exceedingly low doses that do not diminish antinociception nor cause withdrawal in rodents. This unexpected effect appears to arise from 6BN interactions with ligand-free signaling MOR (termed here MOR-μ*) [20].

## 3. Physiological Role and Regulation of Basal MOR Signaling

### 3.1. Regulation and Influence in Pain and Dependence

The ground-state MOR-μ appears to be in an equilibrium with basally active MOR-μ* [20]. While the physiological regulation of the MOR-μ–MOR-μ* equilibrium remains enigmatic, β-arrestin-2 and c-Src appear to be involved, down-regulating G protein coupling while also mediating alternative signaling processes [21,22]. Diverse studies have demonstrated a role of basal MOR signaling in affecting pain perception, for example, counteracting post-surgical pain sensitization [23,24]. Following repeated opioid dosing, for pain therapy or with illicit use, MOR-μ* displays enhanced and sustained basal activity, involved in analgesia and dependence [25,26,27,28,29,30,31,32], which can differ between tissue type and brain regions [31]. Similarly, increased exposure through enkephalin release during withdrawal was shown to enhance MOR basal signaling, thereby ameliorating withdrawal symptoms [28]. In an opioid-dependent state, reversal of MOR-μ* to MOR-μ appears to be slow and MOR-μ* may be a driving force underlying dependence [30]. Dependence can last for days and weeks and compulsive drug-seeking behavior much longer, both elements of opioid use disorder (OUD) [33]. Dependence is characterized by withdrawal symptoms, either spontaneous or elicited by antagonists. The inverse MOR agonists, naloxone and naltrexone, elicit withdrawal even after opioid agonist drugs have been fully removed from the body. The reaction intensity to naloxone appears to correlate with degree of dependence [29] and to be independent of the opioid agonist load, supporting the notion that naloxone acts directly at ligand-free MOR-μ*. Already, four hours after a single large dose of morphine, naloxone elicits withdrawal jumping in mice (acute dependence), whereas the neutral antagonist 6β-naltrexol (6BN) does not [34]. This result indicates that elevated MOR-μ* may already exist in a dependent state after a single morphine dose. Perhaps owing to the complexity of the MOR signaling process, studies on withdrawal symptoms observed with inverse agonists and neutral antagonists have yielded a range of outcomes, with varying conclusions [35]. The combined results indicate that ligand-free MOR-μ* signaling is regulated and has physiological and pharmacological relevance in pain and dependence.

### 3.2. Neutral Antagonists and Inverse Agonists

Basal receptor activity can be revealed with the use of neutral antagonists and inverse agonists [36]. In vitro and in vivo methods have yielded multiple compounds with a range of effects on basal MOR activity, including diverse neutral MOR antagonists [37,38,39,40] and DOR signaling [41,42]. Naltrexone and naloxone act largely as neutral antagonists at spontaneously signaling MOR-μ* under opioid-naive conditions, whereas both turn into inverse MOR-μ* agonists after morphine pretreatment [34]. The same conversion of antagonists into inverse agonists upon agonist exposure has been demonstrated with DOR [43]. Studies on biased agonist ligands have shown that MOR exists in multiple conformations that trigger distinct signaling pathways [16,17,18]. One can, thus, hypothesize that the dependent MOR-μ* state differs from MOR-μ* signaling under opioid-naïve conditions, with naloxone changing from a neutral antagonist to an inverse agonist. In a similar fashion, the antipsychotic pimavanserin acts as an inverse agonist at 5HT2A when coupling to G_αi1_ but as a neutral antagonist with G_αq/11_ [44], suggesting that MOR sensitivity could also depend on the signaling pathway. As observed with opioid agonists, treatment with antagonists also affects the MOR-μ–MOR-μ* equilibrium. Generally, treatment with inverse agonists tends to sensitize MOR and enhance the active state (MOR-μ*) while neutral antagonists favor the MOR-μ ground state [45]. In addition, naloxone turns from a weak inverse agonist at DOR into a partial agonist at DOR upon pretreatment with an inverse agonist of DOR [46]. These studies combined demonstrate that both MOR and DOR change their conformations when bound to distinct antagonist classes, opening opportunities to influence opioid physiology.

### 3.3. Ligand-Free Signaling of Opioid Receptors in Peripheral Nociceptors

Basal MOR and DOR signaling in peripheral nociceptors also plays a role in neuropathic pain. Jeske [47] reviewed the existence of distinct MOR conformations of varying relative abundance as a function of cellular environment and in the periphery compared to the CNS. In peripheral afferent nociceptors, both MOR and DOR maintain a silent status that cannot be readily activated by opioid agonists, possibly because of interactions with GRK2 and β-arrestin [21,24,48], representing, again, a different form of MOR-μ. Inflammatory stimuli lead to activation, both spontaneously to generate basal signaling and restoration of response to agonists for both MOR and DOR. Thus, nociceptive stimuli generate active MOR-μ* as a physiological countermeasure, leading to abatement of neuropathic pain [24,49]. However, if such basal MOR activity fails to be reversed, or is maintained by opioid drug exposure, it can contribute to chronic neuropathic pain and hyperalgesia, supported by the finding that the peripheral antagonist methylnaltrexone prevents opioid tolerance, dependence, and hyperalgesia [48]. The inverse agonist naltrexone blocks MOR-μ* signaling in peripheral nociceptors, but it is also sufficient to activate silent MOR to MOR-μ*, only detectable after naltrexone washout, suggesting a thermodynamic gain via strong binding to MOR-μ* drives the conversion. This result indicates that antagonists also affect the MOR-μ–MOR-μ* equilibrium. In contrast to naltrexone, incubation with 6BN of nociceptors exposed to inflammatory stimuli reverses MOR-μ* back to the MOR-μ ground state [24]—a first indication that 6BN can reverse sustained MOR-μ* signaling (even if tested only at the high concentration of 10 μM in [24]).

## 4. Model of μ Opioid Receptor Signaling

We proposed a receptor model that includes a key role for ligand-free MOR-μ* signaling (Figure 1) [20]. Previous studies demonstrate ligand-free, basal (constitutive) signaling by MOR-μ* [29,30,31,32], thought to exist in equilibrium with resting MOR-μ. It is understood that both MOR-μ and MOR-μ* can exist in multiple conformations and aggregates, supporting different functional pathways. For example, MOR-μ* generated by spontaneous MOR-μ activation could differ from MOR-μ* acutely generated by an agonist and further from sustained MOR-μ* signaling after prolonged opioid stimulation. Each MOR state could couple to different G_α_ subunits or to arrestins [16], also demonstrated for DOR [24]. A better understanding of MOR-μ and MOR-μ* functions and their equilibrium is critical to resolving open questions about the physiology and pharmacology of opioids, including agonists and antagonists. The red arrow in Figure 1 from MOR-μ* to MOR-μ suggests a pathway from the dependent to the opioid-naïve MOR state, raising the question how this retrograde MOR-μ* to MOR-μ conversion is regulated, with broad implications for opioid therapies.

### 4.1. MOR Activation by Agonists

The model in Figure 1 raises the question of which form of MOR elicits signaling, agonist-bound or ligand-free MOR-μ*, or both. Upon binding, an agonist induces a subtle conformational change that overcomes restraints keeping MOR-μ in its ground state, triggering remodeling of the composite receptor aggregate, subsequently, to engage signaling factors, such as G proteins. Each sequential step can cause conformational changes in the receptor, leading to altered agonist affinity to MOR. Diverse evidence supports the hypothesis that some agonists dissociate rapidly after MOR activation. Proposed in Figure 1, agonists, such as etorphine and morphine, appear to bind MOR-μ with high affinity in rodent brains but rapidly dissociate after activation to ligand-free MOR-μ*, which would then carry out the signaling [20,50]. Etorphine’s dissociation half-life in vivo is ~50 s, whereas in vitro, it increases to ~40 min (depending on incubation conditions), presumably because the receptor aggregate and coupling to downstream signaling factors are altered. Moreover, the extremely low antinociceptive EC50 of etorphine (<0.001 mg/kg in rats) occurs at a low fractional MOR occupancy (~2%), accounting, in part, for its extreme potency, as only a small fraction of MOR needs to be occupied [50]. Similar results were obtained for sufentanil, while partial agonists require higher fractional occupancy for EC50 effects [51]. These results suggest that the agonist binds to a high-affinity site, then triggers coupling to signaling proteins that stabilize MOR-μ* further into a conformation with low affinity for the agonist. Thereby, the life-time of MOR-μ* would determine the duration of agonist action, particularly relevant for agonists with rapid elimination from the brain. Such agonists need to occupy only a small fraction of MOR sites to elicit analgesia, contributing to high potency in agonists, such as etorphine and fentanyl, but they display low potency in competing with antagonist MOR binding in vivo [30], consistent with the MOR model in Figure 1. It remains to be resolved whether this process applies to most or all MOR agonists and to different downstream pathways, including arrestin coupling and internalization upon agonist interaction with MOR. More broadly, GPCRs may follow various activation pathways; whether agonist dissociation after activation occurs rapidly upon activation remains to be studied for each GPCR.

The proposed role of MOR-μ*, as the active signaling molecule, has several implications that require attention. Structural analyses have defined MOR as being either in the active state when bound to an agonist or in an inactive state bound to an antagonist. As MOR-μ* is active without any ligand, one should reconsider assigning active and inactive MOR states for structural analyses. An activated MOR-μ* bound to a neutral antagonist could provide further insight into receptor-folding dynamics. In addition, if ligand-free MOR-μ*carries out signaling, the response duration is determined by the MOR-μ* lifetime. Assuming a short duration of agonist stimulation (short elimination half-life of the agonist), duration of analgesia should be similar for full agonists with short half-lives. Implicated in numerous overdose deaths, fentanyl is nearly 100-times more potent than morphine, but also a full agonist compared to morphine and is, therefore, assumed to generate more MOR-μ*, causing lasting respiratory depression, even when the agonist rapidly declines in the brain. Full and lasting activation to MOR-μ*, possibly endowed with distinct signaling pathways, could account for reported cases of complete short-term memory loss associated with bilateral destruction of the hippocampus (opioid-related amnestic syndrome [52]) in survivors of a near-lethal fentanyl overdose. Lastly, placebo effects have similar characteristics compared to morphine, such as duration [53], implicating endorphin peptides that could also dissociate and act via MOR-μ*. Interactions between agonists and antagonists with MOR-μ and MOR-μ* further clarify these processes.

### 4.2. Blocking MOR Activation and Signaling by Antagonists

Following the logic in the MOR model (Figure 1), antagonists can block signaling through two distinct mechanisms: competitively preventing MOR-μ activation to MOR-μ* by agonists or by acting as ‘inverse agonists’ that block ligand-free MOR-μ* signaling, inducing yet another conformational MOR change. Typical MOR antagonists tend to act as full inverse agonists (e.g., BNTX), while naloxone and naltrexone turn into inverse agonists at the elevated basal MOR-μ* state in the dependent state [34], suggesting differences between MOR-μ* states. This finding raises the question whether naloxone and naltrexone also act as inverse agonists at MOR-μ* acutely generated by an agonist. Binding to ligand-free MOR-μ*, inverse agonists are expected to reduce signaling, blocking analgesia and causing withdrawal with equal potency, regardless of the agonist load in the body—this appears to be the case for naloxone and naltrexone [35,54,55]. Minute doses of naloxone (0.05–0.1 mg i.v.) cause perceptible withdrawal in methadone-maintenance patients (typically on 50–100 mg methadone per day), while the antinociceptive IC50 of naltrexone in mice is only 0.007 mg/kg against a large dose of 30 mg/kg morphine [34]—likely a direct effect on ligand-free MOR-μ*, accounting for extreme naltrexone potency. Conversely, even lethal doses of morphine cannot displace ^3^H-naloxone labeling of MOR sites in rat brain as morphine is expected to have low affinity to MOR-μ* [56].

Neutral MOR antagonists are expected to act in a distinct fashion. Belonging to a class of naloxone and naltrexone analogs with reduced C-6 position, imposing a conformational change in the ring structure, the neutral antagonist 6BN is 100-fold less potent than naltrexone in acutely blocking fentanyl antinociception and causing withdrawal in morphine-treated rhesus monkeys [57], only in part accounted for by its slower access to the brain. Rather, we propose that 6BN must compete with the agonist at the MOR-μ ground state to prevent activation since it binds but fails to block ligand-free MOR-μ* signaling. In contrast to naloxone or naltrexone, 6BN no longer causes withdrawal in dependent animals 4 h after injection of morphine when the agonist is largely eliminated, while the dependent state persists. Further, 6BN even blocks naloxone’s effect—expected for a neutral antagonist [29]. These sharp differences between inverse antagonists and neutral antagonists are accounted for by the MOR model in Figure 1. Because of low 6BN potency in both blocking antinociception and causing withdrawal in dependent animals [29,56,57], 6BN is generally not considered to contribute to naltrexone’s effects as its main metabolite. However, 6BN has potent effects on preventing dependence, discussed further below.

### 4.3. Differences between the New MOR Model and Classical GPCR Multistate Models

The main classical multistate GPCR model proposes a ternary complex between agonist, receptor, and G protein, representing the active signaling state [58], with some modification, including a binary complex of receptor and G protein deemed *inactive*. The ternary complex undergoes several conformational steps in the activation process, but any effect on agonist affinity was been considered. Inherent to the MOR model in Figure 1 is an initial conformational change triggered by agonist binding that leads to a substantial reorganization of the receptor complex into a persistent active signaling state, during which the agonist loses affinity and dissociates, generating an *active* binary receptor complex carrying out the signaling. Persistent receptor signaling after dissociation of the agonist has been demonstrated for rhodopsin, suggested to have wider implications for GPCRs [59]. Thereby, ligand-free signaling can occur both with spontaneous activation (basal signaling) or upon agonist activation (these signaling complexes may differ in composition). In the classical model of GPCRs with basal ligand-free signaling, inverse agonists are thought to drive the basally active receptor state back to the resting state, whereas neutral antagonists do not [58]. However, these assumptions appear not to hold for MOR since pretreatment with the inverse agonist naltrexone generates basal MOR signaling in afferent nociceptors, whereas pretreatment with the neutral antagonist 6BN reverses basal activity and even prevents the action of naltrexone—changes in MOR activity detectable after washout of the ligands [24]. Similarly, treatment of DOR-transfected cells with an inverse agonist reduces agonist binding and enhances inverse agonist binding, suggesting conversion of the resting DOR state to a ligand-free active state similar to MOR-μ*, whereas a neutral antagonist does the opposite [46,60]. When ligand-free MOR-μ* emerges after agonist stimulation, an inverse agonist, such as naloxone and naltrexone, appears to bind with high affinity to block signaling activity without converting MOR-μ* back to the resting MOR-μ state. Moreover, the inverse agonist appears to favor the MOR-μ* state, thereby shifting the equilibrium to MOR-μ* (without enabling signaling when bound). In contrast, a neutral antagonist, such as 6BN, also binds to the active state without blocking signaling—the hallmark of neutral antagonism. In addition, we postulate 6BN gradually accelerates conversion back to the ground state, possibly binding there with higher affinity, without affecting acute agonist effects (analgesia). Hence, the classical multistate model stipulating that inverse agonist reverts the active to the silent ground state does not account for a range of observations obtained with MOR agonists and antagonists. This new interpretation of a multi-state MOR model (Figure 1) will require further testing while opening new avenues for developing safer opioid therapies.

## 5. Properties of 6β-Naltrexol (6BN)

### 5.1. 6BN Pharmacology

As a main metabolite of naltrexone in humans (40–50% of the dose) (Figure 2) [55], 6BN has a solid safety record in humans [13], is stable, orally bioavailable in mice (~25%) [29] and in guinea pigs (~30% [61]), and has a half-life of ~12 h in humans [62], rendering 6BN highly ‘druggable’. Further, 6BN’s in vitro binding affinity to MOR, DOR, and KOR (κ opioid receptor) is 1.4, 29, and 2 nM, respectively, similar to naltrexone’s affinity profile (0.5, 7, and 1 nM, respectively; all Ki values) [55,63]. As an inverse MOR agonist, naltrexone blocks opioid analgesia and causes withdrawal in dependent subjects with equally high potency [54,55] so that it can be administered only 1–2 weeks after complete opioid detoxification. In contrast to naltrexone, 6BN blocks analgesia and elicits withdrawal only at much higher doses than naltrexone [20,55,64]. As a result, 6BN as a metabolite was considered not to contribute to naltrexone’s actions in opioid maintenance therapies [65], even though 6BN exceeds naltrexone’s blood levels because of its longer half-life.

Pharmacokinetic parameters of 6BN differ from those of naltrexone. The elimination half-life of 6BN from blood in humans (~12h) is longer than that of naltrexone (~4h) [62], (and references therein). As a result, 6BN blood levels exceed naltrexone levels after naltrexone administration. On the other hand, naltrexone rapidly enters the brain, whereas 6BN enters more slowly [29,61,66,67]. As naltrexone is not metabolized to 6BN in rodents, one can test the peripheral vs. central effects of both compounds separately. In rodents, naltrexone has similar potency in blocking morphine’s effects on nociception and gastrointestinal motility [29,64,66]. On the other hand, 6BN displays a degree of peripheral over central selectivity (5–10-fold), entering the brain slowly via the blood–brain barrier (BBB) [29,64,66], leading to ten-fold higher initial brain than blood concentrations in rodents [61]. Therefore, the contribution of the metabolite 6BN to central effects of naltrexone (analgesia) is considered low. The degree of peripheral selectivity alone cannot account for 6BN’s much lower anti-analgesic potency than naltrexone but also stems from its competitive mechanism at the resting MOR-μ state (Figure 1). Further, 6BN’s peripheral selectivity as a MOR antagonist appears sufficient for potential therapy of opioid-induced bowel dysfunction, including constipation [62], and for preventing hyperalgesia that results, at least in part, from peripheral opioid effects [11].

### 5.2. Potent Suppression of Opioid Dependence with Low-Dose 6BN (LD-6BN)

Oberdick et al. [67] determined that 6BN given together with morphine to juvenile mice prevents subsequent withdrawal jumping, with an EC50 of ~0.03 mg/kg. This result indicated high potency in a centrally mediated effect, unexpected even when accounting for the immature BBB in mice until day 20 post-partum, allowing rapid access of 6BN to the brain. Safa et al. then showed that 6BN co-administered with methadone potently suppresses withdrawal behavior in adult guinea pigs (IC50 ~ 0.01 mg/kg) [61]. Moreover, LD-6BN given s.c. to pregnant guinea pigs together with methadone suppressed withdrawal behavior in the newborn pups, with an ED50 ~ 0.025 mg/kg 6BN [61], without deleterious effects and well below the IC50 against methadone antinociception (estimated ~1.0 mg/kg) [61].

Assuming dependence is largely a centrally mediated mechanism, how can 6BN have potent long-lasting central effects? After high oral doses of 6BN in pregnant guinea pigs, the initial blood/brain 6BN ratio is ~10, but 6BN is rapidly eliminated from blood while it continues to enter and then persist in the brain [61]. As a result, 6BN brain levels exceed blood levels 4 h after dosing following a 1 mg/kg 6BN dose (s.c.) to pregnant guinea pigs. Further, 6BN brain levels remain detectable at ~1 nM in maternal and fetal brains but drop below the threshold of detection in blood at 8 h (~0.2 nM). (Figure 3). Sustained 6BN retention in the brain can be accounted for by high-affinity binding to receptor sites, augmented by retention in a receptor micro-compartment described for diprenorphine [68]. This kinetic model stipulates that the receptor resides in a region of restricted diffusion (e.g., the synaptic cleft) so that a high-affinity ligand rebinds multiple times after dissociation at low receptor occupancy before leaving the micro-compartment. In addition, Porter et al. [55] reported extreme potency for 6BN in the electrically stimulated guinea pig ileum, with a Ki of 94 pM, more potent than naloxone and naltrexone in this assay, suggesting MOR exists in conformations with high affinity for 6BN. Full kinetic analysis of retention in the brain after low 6BN doses (0.02–0.03 mg/kg) and binding to high-affinity sites in mice and guinea pigs is ongoing (Oberdick et al., unpublished).

Extended retention in the brain indicates that 6BN can act centrally, capable of modulating MOR-μ* activity, possibly by accelerating gradual conversion of MOR-μ* back to the resting-state MOR-μ* without substantially blocking MOR-μ* signaling acutely (Figure 2). Moreover, 6BN could favor a MOR-μ ground state thermodynamically because of high binding affinity. As the MOR receptor concentration in the rodent brain is in a range around 10 nM [50,51,68] and assuming a substantial portion of 1 nM 6BN in the brain (Figure 3) is bound to high-affinity sites, we can estimate an approximate receptor occupancy of ~10% or less. At such low MOR occupancy, opioid agonist effects are barely affected while sufficient for 6BN to catalyze the proposed enhanced reversal of MOR-μ* to MOR-μ. The proposed high 6BN potency in reversing opioid dependence, if mediated by MOR-μ* to MOR-μ conversion, could result from both high affinity and low required receptor occupancy over long times, without acutely blocking agonist antinociception. Similar to enhanced potency of agonists (e.g., etorphine) resulting from effective receptor activation at low occupancy [51], low 6BN occupancy can be considered a main catalyst in conveying efficacy as a ‘dependence modulator’, despite slow access to the brain.

While 6BN does not cause withdrawal in an acute morphine-dependence model in mice [26], some withdrawal behavior can be observed at high 6BN doses in a chronic dependence model in mice after intensive morphine treatment when the agonist is eliminated [29], while the withdrawal was less severe than with naloxone. At least two hypotheses can account for this finding: endogenous opioid peptides are upregulated [28] or full occupancy of MOR by high-dose 6BN accelerates the rate of retrograde MOR-μ* to MOR-μ conversion sufficiently to cause withdrawal. The conversion rates between opioid-naïve and dependent states require further study.

### 5.3. Role of 6β-Naltrexol in the Effects of Very-Low-Dose Naltrexone

Because its low potency as an antagonist (64,65], 6BN was not thought to contribute to naltrexone’s effects, even though its affinity to MOR is similar. On the other hand, very-low doses of naltrexone (VLD–naltrexone, defined as 0.001–1.0 mg total oral dose in humans, below the threshold causing withdrawal in dependent subject [69]) have been tested in a variety of indications. For example, VLD–naltrexone is proposed to improve analgesic efficacy, reduce relapse [69,70,71], and facilitate weaning (maximally tolerated dose not causing withdrawal symptoms: 0.25 mg naltrexone [70]), by yet poorly defined mechanisms. Further, VLD–naltrexone reduced drug craving after opioid weaning [70]. While any possible role of 6BN had been discounted, the high 6BN potency in preventing dependence and its accumulation in blood above its parent naltrexone (longer t1/2 (~12h) in humans) [62] makes it the likely active agent in VLD–naltrexone. Low-dose 6BN itself can be titrated to higher doses than naltrexone before withdrawal occurs in dependent subjects, thereby rendering 6BN potentially more effective than what is claimed for VLD naltrexone. It is noted that ultra-low-dose naltrexone (defined as <0.001 mg/kg in animals [69]) was proposed to boost analgesic opioid effects [70]. However, the mechanism remains elusive and is not considered further here in the context of 6BN.

### 5.4. Potential Clinical Uses of 6b-Naltrexol

The unique properties of LD-6BN support its clinical potential as an ‘retrograde addiction modulator’ (Figure 4) in multiple applications, including safety and utility of opioid analgesics and OUD therapies. Further, 6BN acts in three distinct ways as a function of dose [20,61,66]: 1. modulating elements of addiction at low doses; 2. in addition, selectively blocking peripheral opioid effects at intermediate doses; and 3. antagonizing antinociception at high doses. Moreover, intermediate 6BN doses can be selected such that 6BN accumulation in the body reaches levels that blunt central effects upon high-frequency dosing of opioid agonists with shorter half-lives than 6BN, reducing OUD risk [20]. We propose that 6BN is a candidate drug that can improve opioid pain therapy, reduce OUD risk, facilitate opioid-weaning detoxification, treat neonatal opioid withdrawal syndrome [61], and improve the safety of opioid maintenance therapies.

## 6. Discussion

This review assesses the role of ligand-free signaling of MOR as a salient example of numerous GPCRs with similar properties. A main tenet of the model (Figure 1) implies that MOR displays spontaneous basal ligand-free activity (MOR-μ*) and upon activation by agonists, MOR-μ* could account for most or all MOR signaling. In addition, prolonged agonist exposure leads to sustained MOR-μ* activity as one potential driver of dependence, OUD, and other adverse effects. It remains to be determined whether and how MOR-μ* conformations and functions differ under distinct conditions and whether most agonists rapidly dissociate from activated MOR-μ*, leaving the ligand-free receptor to determine the magnitude and duration of activity.

A prominent role of ligand-free MOR-μ* signaling further implies distinct roles for inverse agonists and neutral antagonists. On thermodynamic grounds, ligands can stabilize the MOR receptor conformation towards those with the highest affinity for the ligand, while that process may trigger further remodeling of the receptor aggregate to either initiate signaling or block it (agonists, neutral agonist, and inverse agonist). The proposed novelty of the MOR model in Figure 1 is the hypothesis that a neutral antagonist has the potential to accelerate gradual reversal of MOR-μ* in the dependent state to the resting state—thereby, reversing dependence, captured by the term ‘retrograde addiction modulation’ (Figure 4). Reversal of ligand-free signaling of a two-state receptor-signaling complex by a neutral antagonist is opposite to the assumptions made in classical GPCR models [58]. Retrograde addiction modulators as clinical agents must also be neutral antagonists to allow opioid analgesia and avoid acute withdrawal. These combined properties could lead to a new class of drugs for OUD and pain therapy, exemplified by 6BN and analogs shown to be neutral MOR antagonists [34,37,38,39,72,73].

The MOR model proposed in Figure 1 serves as a guide for further study, according to the motto ‘no model is perfect but some are useful’, potentially applicable to many GPCRs. This review lays out opportunities that could lead to novel means of dealing with the opioid crisis.

## 7. Patents

The following granted patents pertain to 6β-naltrexol and congeners: US8,883,817; 8,748,448; 9,061,024; 10,925,870.

## Figures and Tables

**Figure 1 molecules-27-05826-f001:**
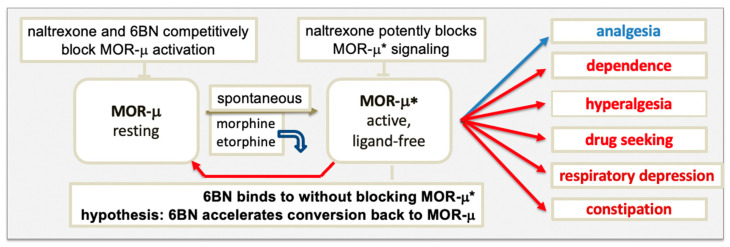
Model of μ opioid receptor (MOR) conformations. MOR-μ* is reversibly activated by both spontaneous conversion and agonists to ligand-free active MOR-μ*. Opioid agonists appear to dissociate from MOR-μ* by losing high-affinity binding. The inverse agonist naltrexone potently blocks ligand-free MOR-μ*, whereas the neutral antagonist 6BN binds to MOR-μ* but does not suppress signaling. Both naltrexone and 6BN block agonist activation competitively at MOR-μ with lower potency. Continued opioid agonist stimulation shifts the equilibrium to persistent MOR-μ* signaling, leading to a dependent state. In contrast to naltrexone, 6BN is proposed to potently accelerate reversal of the equilibrium back to MOR-μ in the opioid-naïve state. Adapted from [20].

**Figure 2 molecules-27-05826-f002:**
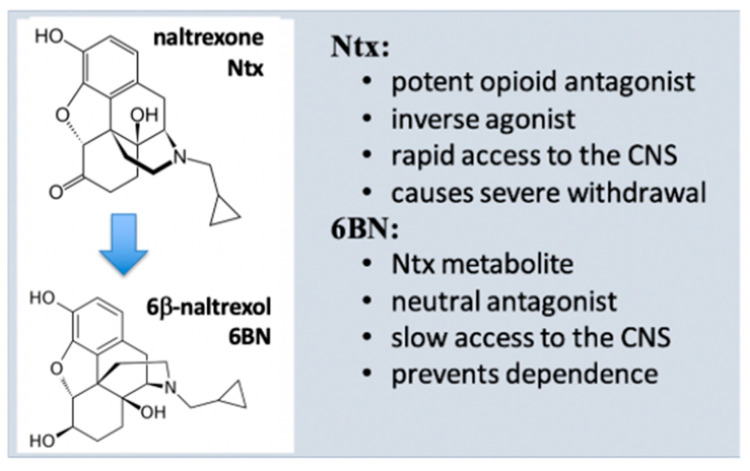
Pharmacological properties of naltrexone and 6β-naltrexol.

**Figure 3 molecules-27-05826-f003:**
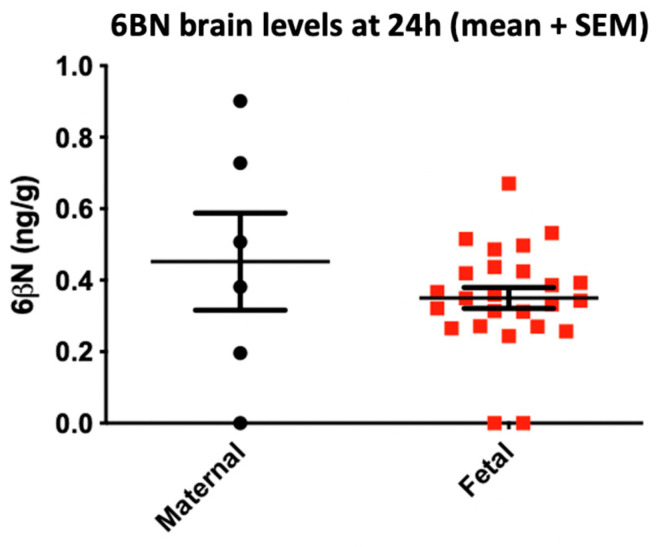
Maternal and fetal brain 6BN levels 24 h after s.c. doses of 1.0 mg/k to pregnant guinea pigs (gestational day 48–51). 6BN brain levels measured with a mass spectrometry assay (described in [61]) remain detectable (~0.5 ng/g, or ~1.5 nM, mean) 24 h after dosing in both dam and fetus. Maternal blood levels decline with a t1/2 of ~3o min in the first 6 h after administration and then become undetectable (~0.1 ng/mL) (performed under an NCATS-Aether CRADA agreement 14112019BC, by Lovelace Biomedical).

**Figure 4 molecules-27-05826-f004:**
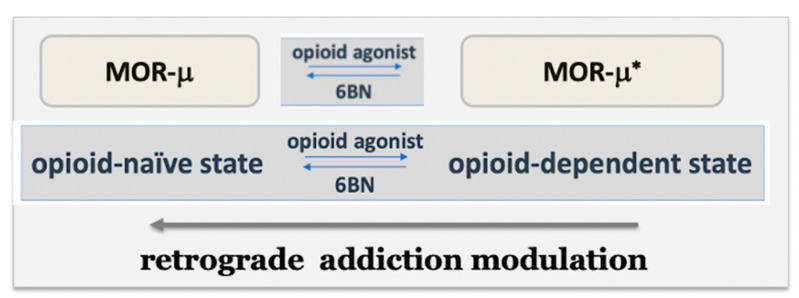
Hypothesis: Accelerating reversal of elevated ligand-free MOR-μ* activity to the resting MOR-μ state, 6BN gradually reverses the opioid-dependent state, directly binding to MOR-μ* without competing with opioid agonists that have low affinity to MOR-μ*. The following term is suggested for this action of 6BN: ‘retrograde addiction modulation’.

## Data Availability

Not applicable.

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
