# Peer review of "Ligand-Free Signaling of G-Protein-Coupled Receptors: Relevance to μ Opioid Receptors in Analgesia and Addiction"

_molecules, 2022, doi:10.3390/molecules27185826_

Round 1
Reviewer 1 Report
In this review article, the authors discussed the basally active state of MOR and the relationship between the pharmacological effects and the mode of action of ligands at the MOR. In general, the basal activities of GPCR, constitutive activities were explained by a so-called two-state receptor model. The model says as follows:
“The revised and extended model (called two-state) includes an explicit isomerization of the receptor first to an active state (R*) before it can couple to the G-protein (Samama et al., 1993). According to this model, constitutive activation has been explained as an alteration of the normal equilibrium between the inactive state (R) and the active state (R*), shifting a higher proportion of receptor molecules in the active R* state. Inverse agonists, previously referred to as negative antagonists such as ICI 118551 for the 2-AR, have a higher affinity for the inactive state R. Therefore, inverse agonists can reverse a constitutively active phenotype of higher basal activity by shifting the equilibrium of the constitutively active receptor back to the inactive state. Neutral antagonists bind with equal affinity to both R and R*. Therefore, neutral antagonists are unable to shift equilibrium and have no effect on the basal activity of constitutively active receptors.” (Pharmacol. Rev. 2005, 57, 147.)
However, the discussion by the authors seems not to be based on the two-state receptor model. When the authors will propose the other theory, the authors should explain their proposal in detail. It would be also important to clarify the differences between their proposal and the well-known theory.
It is also important to define the terms. For example, what does “non-competitively” mean in the sentence “naloxone acts non-competitively at ligand-free MOR-m*(line 100)”? At the ligand-free MOR-m*, there are no ligands. Therefore, I feel that there are neither non-competitive nor competitive interactions.
As a result, it is very hard to correctly understand the assertion indicated by Figure 1.
The authors proposed that the differences of the observed pharmacological effects would stem from the differences of actions between the ligands and the receptor. However, the other many factors such as the ADME, the residence-time of the compound at the target area, and so on would influence the observed pharmacological effects. Therefore, the discussion based on the comparison between the inverse agonists (NTX and NLX) and neutral inverse antagonist (6BN) seems to be extremely simplified. The other information like plasm-concentration, the central-concentration, the duration of action. should be also discussed.
Author Response
Wolfgang Sadee * , John C McKew
Reviewer 1
Review Report (Round 1)
In this review article, the authors discussed the basally active state of MOR and the relationship between the pharmacological effects and the mode of action of ligands at the MOR. In general, the basal activities of GPCR, constitutive activities were explained by a so-called two-state receptor model. The model says as follows:
“The revised and extended model (called two-state) includes an explicit isomerization of the receptor first to an active state (R*) before it can couple to the G-protein (Samama et al., 1993). According to this model, constitutive activation has been explained as an alteration of the normal equilibrium between the inactive state (R) and the active state (R*), shifting a higher proportion of receptor molecules in the active R* state. Inverse agonists, previously referred to as negative antagonists such as ICI 118551 for the 2-AR, have a higher affinity for the inactive state R. Therefore, inverse agonists can reverse a constitutively active phenotype of higher basal activity by shifting the equilibrium of the constitutively active receptor back to the inactive state. Neutral antagonists bind with equal affinity to both R and R*. Therefore, neutral antagonists are unable to shift equilibrium and have no effect on the basal activity of constitutively active receptors.” (Pharmacol. Rev. 2005, 57, 147.)
However, the discussion by the authors seems not to be based on the two-state receptor model. When the authors will propose the other theory, the authors should explain their proposal in detail. It would be also important to clarify the differences between their proposal and the well-known theory.
We appreciate the comments of the reviewer and have included the Pharmacol Rev. reference, and have added more discussion:
“The classical multistate GPCR model proposes that an inverse agonist drives the basally active receptors state back to the resting state, whereas a neutral antagonist does not. However, this assumption does not appear to hold for mu opioid receptors since the inverse agonist naltrexone generates basal MOR signaling in afferent nociceptors whereas the neutral antagonist 6BN reverses basal activity and even prevents the action of naltrexone [48]. Inherent to the MOR model in Figure 1 is an initial conformational change triggered by agonist binding that leads to a substantial reorganization of the receptor complex into a persistent active signaling state, while the agonist looses affinity and dissociates. In the case of MOR-m*, an inverse agonist appears to bind with high affinity to and stabilizes the active state while also blocking signaling. In contrast a neutral antagonist would bind also to the active state but accelerates conversion back to the ground state, presumably binding there with higher affinity. Hence, the classical multistate model does not account for the results obtained with MOR agonists and antagonists.”
Perez, D.M.; Karnik S.S. Multiple signaling states of G-protein-coupled receptors. Pharmacol . Rev, 2005, 57, 147-61. PMID: 15914464
Schafer C.T.; Fay J.F.; Janz J.M.; Farrens D.L. Decay of an active GPCR: Conformational dynamics govern agonist rebinding and persistence of an active, yet empty, receptor state. Proc Natl Acad Sci U S A 2016, 113-11961-11966. PMID: 27702898
Piñeyro G.; Azzi M.; deLéan A.; Schiller P.A.; Bouvier M. Reciprocal regulation of agonist and inverse agonist signaling efficacy upon short-term treatment of the human delta-opioid receptor with an inverse agonist. Mol Pharmacol 2005, 67, 336-48. PMID: 15496503
It is also important to define the terms. For example, what does “non-competitively” mean in the sentence “naloxone acts non-competitively at ligand-free MOR-m*(line 100)”? At the ligand-free MOR-m*, there are no ligands. Therefore, I feel that there are neither non-competitive nor competitive interactions.
As a result, it is very hard to correctly understand the assertion indicated by Figure 1.
We have modified the text to clarify the mode of action.
The authors proposed that the differences of the observed pharmacological effects would stem from the differences of actions between the ligands and the receptor. However, the other many factors such as the ADME, the residence-time of the compound at the target area, and so on would influence the observed pharmacological effects. Therefore, the discussion based on the comparison between the inverse agonists (NTX and NLX) and neutral inverse antagonist (6BN) seems to be extremely simplified. The other information like plasm-concentration, the central-concentration, the duration of action. should be also discussed.
We have added additional text to address these issues, summarizing briefly multiple studies on these topics. A main point is the long retention of 6BN in the brain after low doses, supported by complete time courses after 0.02 mg/kg in guinea pigs and 0.03mg/kg in mice – with and without naloxone to saturate high affinity opioid receptor sites (to be published separately). In this review, we show retention of 6BN in guinea pig brain24h after administration, while blood levels become undetectable at 5-8 h. In rodents, naltrexone is not converted to 6BN, in contrast to the robust conversion in humans, removing confounding factors on their pharmacological effects.
Reviewer 2 Report
It would be informative if the Authors could emphasize that in patients treated for cancer the risk of them developing OUD is really low (along with respiratory depression). Could this phenomenon be related to increased spontaneous activity of the MOR receptor?
Could the action of 6BN on MOR be tissue-specific? Do MORs differ in the level of spontaneous activation depending on tissue type? Agonists are known to produce different levels of stimulation in different brain structures and is this true for inverse agonists/antagonists as well?
It's only a suggestion, but in section 2 I think that it would be better to limit the list of ADRs to those that are most meaningful in a clinical setting. It's a common misconception that respiratory depression is a concern following the medicinal use of opioids. It could be a problem in pain-free patients that abuse opioids. Pain input is a powerful stimulant of the respiratory system and respiratory depression is a very rare occurence.
Author Response
We have appreciated the reviewers’ critique and have addressed all issues raised.
Reviewer 2.
It would be informative if the Authors could emphasize that in patients treated for cancer the risk of them developing OUD is really low (along with respiratory depression). Could this phenomenon be related to increased spontaneous activity of the MOR receptor?
Opioid pain therapy in cancer often does not lead to OUD, now mentioned, whereas dependence, tolerance, constipation, hyperalgesia, cognitive issues, and immune dysfunction can reduce the its clinical utility. Sustained MOR-m* activity could cause or contribute to most of these adverse effects, but we do not want to speculated further on this point as yet.
Could the action of 6BN on MOR be tissue-specific? Do MORs differ in the level of spontaneous activation depending on tissue type? Agonists are known to produce different levels of stimulation in different brain structures and is this true for inverse agonists/antagonists as well?
Ligand-free MOR-m* activity likely differs between cell type, a result of physiological regulation and distinct proteins contained in the receptor aggregates. We have observed different levels of MOR-m* activity between brain regions [31] (added to the text), while in afferent peripheral nociceptors MOR exist in a non-responsive state with no basal activity until exposed to inflammatory stimuli [48], as discussed in detail in the manuscript. Clearly, much more work needs to be done on this issue.
It's only a suggestion, but in section 2 I think that it would be better to limit the list of ADRs to those that are most meaningful in a clinical setting. It's a common misconception that respiratory depression is a concern following the medicinal use of opioids. It could be a problem in pain-free patients that abuse opioids. Pain input is a powerful stimulant of the respiratory system and respiratory depression is a very rare occurrence.
Prevalence of adverse effects depends on clinical circumstances, whether opioids are used medially or under abuse conditions. With increasing length of opioid exposure – medical or illicit – tolerance and hyperalgesia cause increased dose requirements, while respiratory depression and g.i. effects remains largely sensitive – leading to increased risk over time. Such risk may well be less common with medical opioid use, but is still present. We are not sure how to introduce these concepts succinctly into the manuscript.
Reviewer 3 Report
The review entitled Ligand-Free Signaling of G Protein Coupled Receptors: Relevance to m Opioid Receptors in Analgesia and Addiction was focused on the role of ligand-free signaling of a GPCR, the MOR through a literature revisiting. The paper is well written and organized. Moreover, the MOR model could be useful for further study on other GPCRs and on the hypothesis of new therapeutical approaches.
Minor issues:
Abstract: Page 1 line 13: rewrite the sentences, the main verb is missing.
Page 2 line 85: change c-SARC with c-Src
Page 3 line 126: delete s
Page 3 line 136: use only the abbreviation DOR
Page 3 line 138: delete the point between 48 and 49
Page 4 line 155: delete “adapted from [30]”
Page 8 line 352: use symbol for 6b
Page 9 and 10 revised.
References: use “Reference List and Citations Style Guide for MDPI Journals”
Page 10 reference 10: a wrong symbol in the title.
Ref 24 and Reference 23 are the same, please delete
Author Response
Reviewer 3.
We have addressed all issues raised by reviewer 3, seen in the edited marked-up revision.
The review entitled Ligand-Free Signaling of G Protein Coupled Receptors: Relevance to m Opioid Receptors in Analgesia and Addiction was focused on the role of ligand-free signaling of a GPCR, the MOR through a literature revisiting. The paper is well written and organized. Moreover, the MOR model could be useful for further study on other GPCRs and on the hypothesis of new therapeutical approaches.
Minor issues:
Abstract: Page 1 line 13: rewrite the sentences, the main verb is missing.
“signals” is the verb
Page 2 line 85: change c-SARC with c-Src
Page 3 line 126: delete s. (??)
Page 3 line 136: use only the abbreviation DOR
Page 3 line 138: delete the point between 48 and 49
Page 4 line 155: delete “adapted from [30]”
Page 8 line 352: use symbol for 6b. (?...we have used 6BN throughout)
Page 9 and 10 revised. (?)
References: use “Reference List and Citations Style Guide for MDPI Journals”
Page 10 reference 10: a wrong symbol in the title.
Ref 24 and Reference 23 are the same, please delete
Round 2
Reviewer 1 Report
In the revised manuscript, the new section entitled “Differences between the MOR model (Fig. 1) and classical GPCR multistate models” is added. The new section describes not recently authorized theory but the authors’ proposal. Indeed, the notion that naloxone and naltrexone are inverse agonist and that 6BN is a neutral antagonist is shown in the last part of the section. However, any appropriate references are cited. The authors said that inverse agonists (naloxone and naltrexone) bind to the active sate of MOR and stabilize it, and further to block signaling. What does it mean? In general, agonists bind to the receptors in active state and stabilize them to maintain or amplify signaling. Does the MOR in the active state bound to agonists and that bound to invers agonists lead to the MOR with different conformation? The description that a neutral antagonist such as 6BN also binds to the active state but without blocking signaling is coincident with the definition of a neutral antagonist described by the reference 58. However, the authors proposed that 6BN accelerate conversion back to the ground state. This function seems to be inverse agonism described be the reference 58. When the authors propose a new theory, it should be described as a main theme in an appropriate article.
There is no reply concerning a previous comment “It is also important to define the terms. For example, what does “non-competitively” mean in the sentence “naloxone acts non-competitively at ligand-free MOR-m*(line 100)”? At the ligand-free MOR-m*, there are no ligand. Therefore, I feel that there are neither non-competitive nor competitive interactions.”
Therefore, I cannot recommend the revised to be published.
Author Response
Reviewer 1
In the revised manuscript, the new section entitled “Differences between the MOR model (Fig. 1) and classical GPCR multiistate models” is added. The new section describes not recently authorized theory but the authors’ proposal. Indeed, the notion that naloxone and naltrexone are inverse agonist and that 6BN is a neutral antagonist is shown in the last part of the section. However, any appropriate references are cited. The authors said that inverse agonists (naloxone and naltrexone) bind to the active sate of MOR and stabilize it, and further to block signaling. What does it mean? In general, agonists bind to the receptors in active state and stabilize them to maintain or amplify signaling. Does the MOR in the active state bound to agonists and that bound to invers agonists lead to the MOR with different conformation? The description that a neutral antagonist such as 6BN also binds to the active state but without blocking signaling is coincident with the definition of a neutral antagonist described by the reference 58. However, the authors proposed that 6BN accelerate conversion back to the ground state. This function seems to be inverse agonism described be the reference 58. When the authors propose a new theory, it should be described as a main theme in an appropriate article.
We have modified the text to clarify these issues and have edited the paragraph as follows.
3.3. Differences between the MOR model (Fig. 1) and classical GPCR multistate models
The main classical multistate GPCR model proposes a ternary complex between agonist, receptor, and G protein representing the active signaling state [58], with some modification including a binary complex of receptor and G protein deemed inactive. The ternary complex undergoes several conformational steps in the activation process, but any effect on agonist affinity had not been considered. Inherent to the MOR model in Figure 1 is an initial conformational change triggered by agonist binding that leads to a substantial reorganization of the receptor complex into a persistent active signaling state, during which the agonist loses affinity and dissociates, generating an active binary receptor complex carrying out the signaling. Persistent receptor signaling after dissociation of the agonist has been demonstrated for rhodopsin, suggested to have wider implications for GPCRs [59]. Thereby, ligand-free signaling can occur both with spontaneous activation (basal signaling) or upon agonist activation (these signaling complexes may differ in composition). In the classical model of GPCRs with basal ligand-free signaling, inverse agonists are thought to drive the basally active receptor state back to the resting state, whereas neutral antagonists do not [58]. However, these assumptions appear not to hold for MOR since pretreatment with the inverse agonist naltrexone generates basal MOR signaling in afferent nociceptors whereas pretreatment with the neutral antagonist 6BN reverses basal activity and even prevents the action of naltrexone – changes in MOR activity detectable after washout of the ligands [24]. Similarly, treatment of DOR transfected cells with an inverse agonist reduces agonist binding and enhances inverse agonist binding, suggesting conversion of the resting DOR state to a ligand-free active state similar to MOR-m*, whereas a neutral antagonist does the opposite [60]. When ligand-free MOR-m* emerges after agonist stimulation, an inverse agonist such as naloxone and naltrexone appears to bind with high affinity to block signaling activity without converting MOR-m* back to the resting MOR-m state. Moreover, the inverse agonist appears to favor the MOR-m* state, thereby, shifting the equilibrium to MOR-m* (without enabling signaling when bound). In contrast, a neutral antagonist such as 6BN also binds to the active state without blocking signaling – the hallmark of neutral antagonism. In addition, we postulate 6BN gradually accelerate conversion back to the ground state, possibly binding there with higher affinity, without affecting acute agonist effects (analgesia). Hence, the classical multi-state model stipulating that inverse agonist revert the active to the silent ground state does not account for a range of observations obtained with MOR agonists and antagonists. This new interpretation of a multi-state MOR model (Figure 1) will require further testing while opening new avenues for developing safer opioid therapies.
There is no reply concerning a previous comment “It is also important to define the terms. For example, what does “non-competitively” mean in the sentence “naloxone acts non-competitively at ligand-free MOR-m*(line 100)”? At the ligand-free MOR-m*, there are no ligand. Therefore, I feel that there are neither non-competitive nor competitive interactions.”
We had addressed this issue but overlooked the mention of ‘non-competitive’ on line 109 and 2 other locations. The revised sentence in line 109 now reads: “…supporting the notion that naloxone acts directly at ligand-free MOR-m*.”
Further minor edits have been carried out to clarify text and avoid any errors.